# Techno-biofunctionality of mangostin extract-loaded virgin coconut oil nanoemulsion and nanoemulgel

**Chatchai Sungpud**[1], **Worawan Panpipat**[1]*, **Manat Chaijan**[1], **Attawadee Sae Yoon**[2]

**1** Food Technology and Innovation Research Center of Excellence, Department of Agro-Industry, School of Agricultural Technology, Walailak University, Nakhon Si Thammarat, Thailand, **2** Drug and Cosmetics Excellence Center, School of Pharmacy, Walailak University, Nakhon Si Thammarat, Thailand

* pworawan@wu.ac.th

**Data Availability Statement:** All relevant data are within the manuscript and its Supporting Information files.

## Abstract

Techno-biofunctional characteristics of nanoemulsion and (nano)emulgel loaded with mangostin extracts were elucidated. Crude mangostins from mangosteen peels recovered by virgin coconut oil (VCO), mixed VCO and propylene glycol (PG), and pure PG were used. The extracts were loaded in the dispersed phase in the presence of mixed surfactants (Tween20/Span20) with a varying hydrophilic-lipophilic balance (HLB) from 10.2 to 15.1. Results showed that globular and uniformly distributed droplets of the nanoemulsion were observed. The small particle sizes (typically 18–62 nm) with the zeta potential of -39 to -54.5 mV were obtained when mixed emulsifiers with HLB values of 12.6 and 15.1 were employed. With HLB values of 12.6 and 15.1, nanoemulsions loaded with mangostin extracts prepared with mixed VCO-PG and pure PG-based extracts showed approximately a 2 to 3-fold lower droplet size diameter when compared with the VCO-based extract. For the stability test, all nanoemulsions were stable over three freeze-thaw cycles with some changes in pH, zeta potential, and droplet size. The DPPH$^\bullet$ scavenging activity, H$_2$O$_2$ scavenging activity, reducing power and antibacterial activities (*E. coli* and *S. aureus*) of the nanoemulsions were greater than their corresponding bulk extracts. Nanoemulgels produced by embedding the nanoemulsions in a hydrogel matrix was homogeneous and creamy yellow-white in appearance. The nanoemulgels had a higher mangostin release (87–92%) than their normal emulgels (74–78%). Therefore, this study presented the feasibility of nanoemulsions and nanoemulgels loaded with mangostin extracts as a promising delivery system for bioactive polyphenol in food supplements, pharmaceuticals and cosmetics.

## Introduction

Bioactive polyphenols (e.g. α-mangostin, β-mangostin and γ-mangostin) recovered from mangosteen (*Garcinia mangostana* L.) pericarp have been used worldwide in traditional medicine [1] as an antioxidant [2] and antimicrobial agent [3]. Generally, bioactive polyphenol from

**Funding:** The authors gratefully acknowledge the financial supports from Walailak University (Grant No. 22/2562) and Nakhon Si Thammarat Rajabhat University to CS. This research was partially supported by the New Strategic Research (P2P) project, Walailak University, Thailand to WP.

**Competing interests:** The authors have declared that no competing interests exist.

mangosteen peels was extracted by organic solvents [4], which were toxic and presented an environmental concern. Also, solvent removal was needed. To overcome this problem, an attempt was made to use alternative bio-based solvents in particular virgin coconut oil (VCO) together with propylene glycol (PG) for maximizing polyphenol recovery from mangosteen pericarp [5]. Since PG has found a steadily increasing usage as a constituent of the bases of many different pharmaceutical preparations and cosmetic articles, certain studies reported the various allergic, or possibly allergic dermatoses because of excessive dehydration of the skin after PG pallication [6]. Therefore, the combination of PG with other solvents such as vegetable oil can be an easy strategy to increase its safety.

Plant-based oils have been given more attention for the development of natural and eco-friendly cosmetics [7] with antimicrobial activity [8]. Among vegetable oils, coconut oil was the candidate to be used for pharmaceutical and cosmetic purposes because of its wide range of biofunctional properties. Coconut oil was traditionally used as hair and skin moisturizers in some communities [9, 10]. Moreover, the fatty acids composed in coconut oil (e.g. lauric acid) provided high antimicrobial potency by disrupting cell membranes [11, 12]. Therefore, the utilization of coconut oil together with a classical PG solvent as a bio-based extractant for recovering mangosteen peel polyphenols with duo-functionality is of great interest in the present study. Moreover, both coconut oil and PG mangosteen peel-based extracts can be applied directly in many topical formulations [13] without any solvent removal.

The direct incorporation of phenolic-rich bio-oil in topical cosmetics is limited by several technological challenges, which concern their adequate dispersion in the matrix, the control of their interaction with the other ingredients, as well as the preservation of their activity for the required time [14, 15]. To achieve effective topical application, the mangosteen oil based phenolics should be in a form where rapid diffusion through the skin is reached. Formulation of polyphenols into oil in water (o/w) emulsions can be used to enhance their bioactivity and bioavailability [16]. Furthermore, encapsulation of mangosteen phenolics into emulsions can retard the phenolic degradation and increase the bioavailability of the extract [17]. However, the skin diffusion rate, bioavailability and bioactivity of encapsulated polyphenols in o/w emulsions depend on the oil droplet size. As a result, nanoemulsions should play a greater role and receive much more attention as a bioactive polyphenol delivery system.

Nanoemulsions are transparent colloidal dispersions having the average droplet size of 20–500 nm [18] with a long term physical stability and a wide range of applicability in food, drugs, cosmetics, and skin care products [19, 20, 21, 22, 23, 24, 25]. However, the formation and properties of nanoemulsions are largely influenced by production method, viscosity of the oil dispersive phase, type of surfactants and their hydrophilic lipophilic balance (HLB) values [25, 26, 27]. To gain an excellent mangosteen polyphenolic loaded nanoemulsion, factors affecting the nanoemulsion characteristics should be intensively investigated. Nevertheless, mangosteen phenolic loaded nanoemulsions may have limited use in transdermal delivery because of its low viscosity. Several literatures have reported the better effect of emulgels for controlled release, skin absorption rate, and bioavailability [28, 29, 30]. The incorporation of the gel-forming agent in the water phase can change a classical emulsion into an emulgel with desirable properties [31, 32, 33]. For nanoemulgels, they can be made by incorporating nanoemulsions into a gel matrix.

This study aimed to evaluate the feasibility of nanoemulsions and their nanoemulgels as a carrier for targeting mangosteen phenolics to transdermal cells. The effects of the different types of oil-loading solvents (VCO, mixed VCO-PG, and PG) and HLB values of mixed surfactants (Tween20 and Span20) of nanoemulsions on the physical properties, antioxidative capacities, antibacterial properties, and its freeze-thaw stability were investigated. Furthermore, the physical quality and appurtenance of nanoemulgels compared to emulgels were analyzed. The

mangostins released from nanoemulgels were also determined by the *in-vitro* a Franz diffusion cell test for evaluating its potential topical application.

## Material and methods

### VCO preparation

The fermentation method was used for VCO preparation [5]. In brief, the coconut milk of a mature coconut from the tall tree cultivar *Cocos nucifera* Linn was stored at -18 ˚C for 2 h. The upper cream layer obtained was subjected to incubation at 35 ˚C for 24 h in a glass bowl. Thereafter, the upper oil phase was collected and dried using a rotary evaporator (Büchi, Rota-vapor R-124, Flawil, Switzerland) at 50–60 ˚C (100 mbar) for 15 min until the moisture content was less than 0.2%.

### Preparation of mangostin extracts

The pericarp of fresh dark reddish-purple mature fruits of the *Garcinia mangostana* Linn from a local farm in Nakhon Si Thammarat, Thailand was dried at 50˚C and ground into powder using the Cyclotec™ 1093 Sample Mill (Foss Tecator Co., Hillerod, Denmark). The fine powder was passed through 100 mm sieve meshes and subjected to extraction by using an ultrasound-assisted extraction (UAE) method. Mangosteen peel powder of 13.3 g was mixed with 100 mL (1:7.5 g/mL) of 3 different solvents (VCO, PG (Sigma Aldrich Co., St. Louis, MO, USA), and a mixture of VCO-PG at 1:1), and then subjected to an ultrasonic probe (13 mm diameter cylindrical titanium alloy head). Solutions were sonicated using Sonics vibra cell VCX500 (USA) at 20 kHz, 500 Watts, and 55% amplitude for 7 min with pulse on 30 sec and off 5 sec in an ice bath. Thereafter, the mixture was centrifuged at 5,000 rpm for 10 min. The resulted supernatants were referred to as bulk mangostin extracts. The α-mangostin and γ-mangostin contents in the extracts were quantified by reverse phase high performance liquid chromatography (HPLC) [5].

### Determination of α-mangostin and γ-mangostin contents

The reverse phase HPLC system (Prominence-I LC-2030C 3D Plus, Shimadzu, Kyoto, Japan) coupled with a Shim-pack GIST C-18 column (4.6 × 150 mm, 5 μm size) and a photodiode array detector was used to identify γ-mangostin and α-mangostin contents in the samples according to Sungpud *et al.* [5].

### Preparation of nanoemulsions loaded with mangostin extracts

Three different mangostin extracts with the same total phenolic content were used as the dispersed phase for nanoemulsion preparation. The dispersed phase (10%) was mixed with the aqueous continuous phase containing 10% mixed emulsifiers with three different HLB values (10.2, 12.6 and 15.1). Mixed emulsifiers were prepared by mixing Span 20 (Sigma Aldrich, HLB = 8.6) and Tween 20 (Sigma Aldrich, HLB = 16.7) at ratios of 1:4, 1:1, and 4:1 (w/w) to obtain HLB values of 10.2, 12.6, and 15.1, respectively. Mangostin extract-loaded coarse emulsions were firstly prepared using an IKA® homogenizer (Model T25 digital Ultra-Turrax®, Staufen, Germany) at 15,000 rpm for 5 min in an ice bath. For nanoemulsion preparation, the resulting coarse emulsions were subjected to an Ultrasonic Processor (VCX750 Vibra-Cell™, Sonics & Materials Inc., Newtown, CT, USA) at 20 kHz, 750 Watts, and an amplitude of 70% for 10 min at pulse on 30 sec and off 5 sec in an ice bath. The α-mangostin contents in the VCO, VCO-PG, and PG-based nanoemulsions were 31.62, 35.69, and 40.38 mg/100 g emulsion, respectively. The γ-mangostin contents in the VCO,

VCO-PG, and PG-based nanoemulsions were 13.75, 14.31, and 15.67 mg/100 g emulsion, respectively. Nanoemulsions were equilibrated at 25 ˚C for 60 min before the analysis of pH, particle size diameter and distribution, zeta potential, oil droplet morphology, antioxidant activity, antimicrobial activity, and freeze-thaw stability. The nanoemulsions that exhibited the greatest techno-functional properties were selected for production of nanoemulgels.

## Physico-chemical property of nanoemulsions loaded with mangostin extracts

**Determination of pH.**   The pH of nanoemulsions loaded with mangostin extracts was measured by a pH meter (Mettler Toledo, Columbus, OH, USA).

**Determination of droplet size diameter (mean particle diameter).**   The droplet size was measured using the Zetasizer Nano-ZS90 (Malvern Instruments Ltd., Malvern, Worcestershire, UK). The He-Ne laser emission (633 nm) and the scattering angle of 173˚ (25˚C) were used. The refractive index for VCO, PG and deionized water (25˚C) was set at 1.449, 1.332, and 1.330, respectively. The dilution (100-fold) was made with deionized water prior to measurement. The intensity average hydrodynamic diameter was used to quantify the average droplet size. The Z-average diameter was reported as the average size diameter. The polydispersity index (PDI) value was reported to identify the quality of the droplet size distribution.

**Determination of zeta potential.**   The zeta potentials of nanoemulsions (100-fold dilution with distilled water) were evaluated by quantifying the electrophoretic mobility at an angle of 90˚ using the Zetasizer Nano-ZS90.

**Determination of emulsions morphology.**   The particle morphology of the nanoemulsions loaded with mangostin extracts with the smallest particle size was done using a Leica SP5II confocal laser scanning microscope (CLSM) (Leica, Heidelberg, Germany). A 100-fold dilution with distilled water was made for the freshly made nanoemulsions (100 μL) prior to mixing well with 20 μL 1% methanolic Nile red dye (Sigma Aldrich) solution. After incubation at room temperature (27–29˚C) for 5 min, 20 μL of stained sample was smeared on the microscope slide. The CLSM was done in the fluorescence mode using a Helium Neon Red laser (HeNe-R). The excitation and emission wavelengths were set at 543 nm and 600 nm, respectively. A magnification of 1,100x was used.

## Determination of nanoemulsion stability

All formulations of nanoemulsions loaded with mangostin extracts were determined for the freeze-thaw stability [34]. The samples were stored at -20˚C for 48 h and subsequently thawed up to 25˚C at room temperature. The freeze-thaw was repeated for three cycles. The stability indices including pH, particle size distribution and zeta potential were analyzed.

## Determination of antioxidant activities

**DPPH$^\bullet$ scavenging activity.**   The scavenging activity of the stable 1,1-diphenyl-2-picrylhydrazyl free radical (DPPH$^\bullet$) was evaluated according to Shimada *et al.* [35]. Briefly, 1 mL of nanoemulsions loaded with mangostin extracts (5–50 mg/mL) was mixed well with 1 mL 0.2 mM methanolic DPPH$^\bullet$ solution. The mixture was shaken vigorously and incubated in the dark at room temperature for 30 min. Then the absorbance was read at 517 nm against a blank using a Libra S22 UV-Visible spectrophotometer (Biochrom, Cambridge, England). The control was prepared with the same procedure but distilled water was used instead of the sample.

DPPH$^{\bullet}$ scavenging activity was calculated using the following formula:

$$\text{DPPH radical scavenging activity } (\%) = \left[\frac{A0 - A1}{A0}\right] \times 100 \qquad (1)$$

where A0 was the absorbance of the control and A1 was the absorbance of the sample.

The half-maximal effective concentration (EC$_{50}$; mg/mL) value was reported based on the amount of the extracts which was necessary to scavenge the initial DPPH$^{\bullet}$ concentration by 50%. The α-tocopherol was used as a positive control.

**Hydrogen peroxide (H$_2$O$_2$) scavenging activity.** The H$_2$O$_2$ scavenging activity was measured by the method of Mukhopadhyay *et al*. [36]. To begin with, 20 μL of ferrous ammonium sulphate (1 mM) was added to a series of 96 well plates. Then, 120 μL of different concentrations of nanoemulsions loaded with mangostin extracts with serial dilutions by deionized water were added. Next, 50 μL of 25 mM H$_2$O$_2$ was added and thoroughly mixed using a vibration mode. Thereafter, the solution was allowed to stand at room temperature in the dark for 5 min. After incubation, 120 μL of 1 mM 1,10-phenanthroline was added to each well, mixed, and incubated for 10 min at room temperature. Finally, absorbance was taken at 510 nm through a BMG Labtech-SpectrostarNano plate reader (NC, USA). The control solution was prepared with the same procedure but distilled water was used instead of the sample. The blank solution was also prepared with the same protocol as the control but distilled water was added instead of H$_2$O$_2$. A reagent blank containing only 1,10-phenanthroline was prepared and the absorbance of this tube was subtracted from all tubes. Ascorbic acid was used as a positive control. The H$_2$O$_2$ scavenging activity was calculated using the following formula:

$$\text{Hydrogen peroxide scavenging activity } (\%) = \left[\frac{A0 - A1}{A0}\right] \times 100 \qquad (2)$$

where A0 was the absorbance of the blank-control and A1 was the absorbance of the blank-sample.

The EC$_{50}$ (mg/mL) value was also reported.

**Reducing power.** The reducing power was determined by modified Prussian blue assay [37]. Nanoemulsions loaded with mangostin extracts were 10-fold diluted with a 0.2 M phosphate buffer, pH 6.6. A 70 μL sample solution was mixed with 35 μL 1% (w/v) potassium ferricyanide, and the mixture was incubated at 50°C for 20 min. Then, 135 μL distilled water, 33 μL 10% (w/v) trichloroacetic acid, and 27 μL 0.1% (w/v) ferric chloride were added to the reaction medium and incubated for 10 min at room temperature. Then, absorbance was measured at 700 nm against blank (without the nanoemulsions loaded with mangostin extracts). Ascorbic acid (0–100 μM) was used as a standard. The reducing power was calculated and expressed as milligrams of ascorbic acid equivalents (AAE) per 100 g sample.

**Evaluation of antibacterial activity.** The minimal inhibitory concentration (MIC) of the individual nanoemulsion loaded with mangostin extracts against the foodborne pathogen (*Escherichia coli*) and the major acne bacteria (*Staphylococcus aureus*) were determined by the agar dilution method [38] with some modifications. Sterile test tubes (20 mLX) containing unsolidified tryptic soy agar with 0.6% yeast extract (TSAYE) and Mueller-Hinton agar (MHA) were prepared. Each nanoemulsion loaded mangostin extracts was diluted in serial two folds with dimethyl sulfoxide (DMSO) to obtain the mangostin concentration of 25 to 0.39 mg/mL. The corresponding nanoemulsion loaded mangostin extracts (5 mL) was pipetted to the previous media test tube. Then, 100 μL suspensions (4.0 log cfu/mL) of tested bacteria were added to each tube. The tubes containing test bacteria, nanoemulsion loaded mangostin extracts, and medium agar (TSAYE and MHA) were mixed for 10 s prior to solidification in

plates at room temperature. TSAYE and MHA with ampicillin (ranging from 125 to 1.95 μg/ mL) and DMSO were used as a positive and a negative control, respectively. Finally, plates were incubated at 37 ˚C for 24 h. The MIC, defined as the lowest serial level of sample inhibiting the visible growth of test microorganism on the agar plate, was reported.

## Formulation of emulgel and nanoemulgel

The composition of emulgels and nanoemulgels loaded with mangostin extracts is shown in Table 1. The emulgels and nanoemulgels loaded with mangostin extracts were prepared by mixing the emulsions or nanoemulsions loaded with mangostin extracts in the Component A. The mixture was then heated up to 50 ˚C in a temperature controlled water bath (Memmert, Schwabach, Germany) and stirred at 1,500 rpm by a mechanical stirrer (Dragon lab 22CZ41, Beijing, China). Then, Component B was added into the mixture. The homogeneous mixture was also continuously mixed during cooling to 35 ˚C. Finally, Component C was added into the mixture and then further stirred until the homogeneous gel was obtained. The α-mangostin contents in the VCO, VCO-PG, and PG-based (nano) emulgels were at 28.82, 32.53, and 36.81 mg/100 gel, respectively. The γ-mangostin contents in the VCO, VCO-PG, and PG-based (nano) emulgels were at 12.53, 13.04, and 14.28 mg/100 gel, respectively.

**Physical and chemical characterizations of emulgel and nanoemulgel.** The prepared emulgels and nanoemulgels loaded with mangostin extracts were inspected visually for their clarity, glitter, homogeneity, color, and viscosity. The pH values of 1% aqueous solutions of the samples were measured by a pH meter (Mettler Toledo). The color of the gels was determined using a colorimeter (HunterLab, Colorflex, Hunter Associates Laboratory, VA, USA). CIE $L^*$, $a^*$, and $b^*$ values were recorded. Viscosity was analyzed at 25˚C using a rheometer [30] (HAAKE Mars 60, ThermoFisher Scientific, Karlsruhe, Germany) equipped with a parallel steel lower plate TM-35 and rotor TM-35.

**In vitro mangostins release.** Mangostin release was measured according to the method of Ambala and Vemula [30] with slight modifications. A Franz diffusion cell with effective diffusion area of 1 cm² and 12 mL cell volume was used. Emulgels and nanoemulgels (500 mg) were applied onto the surface of a cellophane membrane evenly. The membrane was clamped up between the donor and the receptor chamber of the diffusion cell. The receptor chamber was filled with freshly prepared 0.1 M phosphate buffer (pH 5.5) containing 0.9% normal

**Table 1. Composition of nanoemulgels loaded with mangostin extracts.**

| Composition | Content (%) |
|---|---|
| **Phase A** | |
| 1. Mangostin extract-loaded emulsion | 91.15 |
| 2. Allontoin | 0.5 |
| 3. Oligosacharide GGF | 1.0 |
| 4. Betaine | 2.0 |
| 5. Glycerol | 2.0 |
| 6. Ascorbic acid | 0.05 |
| **Phase B** | |
| Ammonium acryloyldimethyltaurate/VP | 1.5 |
| **Phase C** | |
| 1. Turmeric oleoresin | 1.0 |
| 2. DMDM hydration | 0.5 |
| 3. Fragrance | 0.3 |

saline to solubilize the sample. The receptor chamber was stirred by a magnetic stirrer (Logan Instruments Corp, VTC 300, NJ, USA) at 37 ˚C for 24 h. The samples were collected and analyzed for α-mangostin and γ-mangostin contents. A percentage of mangostin release was calculated using the following formula:

$$Mangostin\ release\ (\%) = \frac{Mangostins\ content\ in\ receptor\ chamber}{Mangostins\ content\ in\ donor} \times 100 \qquad (3)$$

## Statistical analysis

All experiments were carried out in triplicates and results were expressed as mean values with standard deviation (±SD) of three replicates. One-way analysis of variance (ANOVA) and Duncan's new multiple range test (DMRT) were carried out to determine significant differences ($p < 0.05$) between means by using the SPSS statistical software package (SPSS, version 20.0) (SPSS Inc., Chicago, IL, USA).

## Results and discussion

### Physico-chemical property of nanoemulsions loaded with mangostin extracts

**Droplet size diameter and distribution.** The greater physical stability of nanoemulsions was observed compared to course emulsions due to a reduction in droplet size [39]. The effects of the mixed surfactants (hydrophilic surfactant, Tween 20 and lipophilic surfactant, Span 20) with different HLB values on the droplet size diameter and distribution of the nanoemulsions loaded with mangostin extracts prepared by three dispersive phases including mangostin extract based-VCO, -mixed VCO-PG, and -PG are shown in Table 2. All obtained samples can be characterized as nanoemulsions because their median particle size was less than 500 nm (Table 2). McClements and Rao [40] stated that mixed emulsifiers are typically superior to a single emulsifier in preventing aggregation. The smaller droplets can be obtained when

**Table 2. Effect of extractant and hydrophilic-lipophilic balance (HLB) of surfactant on droplet size diameter and zeta potential of nanoemulsions loaded with mangostin extracts.**

| Extractant | HLB value of mixed surfactant (Span20:Tween20) | Droplet size (nm) | Polydispersity index (PDI) | Zeta potential (mV) |
|---|---|---|---|---|
| VCO | 10.2 | 191.00 ± 2.65[e] | 0.37 ± 0.05[d] | -58.07 ± 0.42[a] |
| | 12.6 | 62.00 ± 1.00[d] | 0.23 ± 0.01[b] | -39.40 ± 0.30[d] |
| | 15.1 | 61.67 ± 2.08[d] | 0.31 ± 0.05[c] | -53.43 ± 0.93[b] |
| Mixed VCO-PG | 10.2 | 279.00 ± 3.61[f] | 0.24 ± 0.01[b] | -53.60 ± 1.11[b] |
| | 12.6 | 35.33 ± 0.58[c] | 0.17 ± 0.01[a] | -39.83 ± 1.72[d] |
| | 15.1 | 28.67 ± 0.58[bc] | 0.37 ± 0.01[d] | -48.50 ± 2.27[c] |
| PG | 10.2 | 394.67 ± 16.29[g] | 0.18 ± 0.01[a] | -49.83 ± 0.80[c] |
| | 12.6 | 22.00 ± 2.00[ab] | 0.34 ± 0.01[cd] | -41.50 ± 1.45[d] |
| | 15.1 | 18.00 ± 2.65[a] | 0.34 ± 0.04[cd] | -54.53 ± 0.55[b] |

Values are given as mean±standard deviation from triplicate determinations.

Different letters in the same column indicate significant differences (p < 0.05).

VCO = virgin coconut oil

PG = propylene glycol

lipophilic and hydrophilic surfactants were applied simultaneously using high energy approaches. The small particle sizes (<100 nm) were observed when the mixed emulsifiers with HLB values of 12.6 and 15.1 were used. Larger median particle sizes (191–395 nm) were obtained when the mixed surfactants with an HLB value of 10.2 was used. This was in agreement with Peshkovsky *et al.* [41], who reported on the effects of HLB values, made by mixing Tween 80 and Span 80, on the particle size of soybean oil nanoemulsions. The increase in the HLB values from 9 to 13 and from 13 to 15 led to a decrease and then an increase in particle size. A steady decrease in particle size of fish oil nanoemulsions was found with increasing the HLB value from 9 to 12 and then remaining constant at higher HLB values [42]. From the results, the smallest droplet size diameter (18–62 nm) was found in nanoemulsions prepared using surfactants with HLB values of 12.6 and 15.1 ($p < 0.05$), whereas the largest droplet size diameter (191–395 nm) was obtained in nanoemulsions prepared using surfactants with an HLB value of 10.2 ($p < 0.05$). The molecular structure and HLB value are important for the selection of emulsifiers to enable the formation and stabilization of the emulsions [43]. The variation in the relative solubility of non-ionic surfactants in dispersed and continuous phases at different HLB values may be correlated with the different droplet sizes [42]. With a higher HLB value, the head part of a non-ionic surfactant is highly hydrated and solubilized in the aqueous phase leading to the smaller particle size of the resulting droplets. With a lower HLB value, the head group is dehydrated and solubilized in the lipid phase leading to the larger particle size of the resulting droplets. Furthermore, the surfactant molecule aligned at the interphase also plays an important role on particle size [42].

The difference in particle size diameter of the nanoemulsions was also affected by the mangostin extracts dissolving medium used at the dispersed phase. At the similar HLB value of 10.2, the smallest particle size diameter was detected in the mangostin extract-VCO-made emulsions (191 nm) followed by the mangostin extract-mixed VCO-PG-made- and PG-made emulsions, respectively (Table 2). However, the lowest particle size diameter was obtained in the mangostin extract-PG-based nanoemulsions prepared with an HLB value of 12.6 ($p < 0.05$). The mangostin extract-based VCO nanoemulsions showed the largest droplet size ($p < 0.05$). A similar trend was obtained in the nanoemulsions with an HLB value of 15.1 (Table 2). The different effects of the dispersed phase on the droplet size might be governed by its viscosity. The more viscous the dispersed phase the more difficult the particle size reduction. Mohammed *et al.* [44] found that the average droplet size increased with the increasing of the dispersed phase viscosity. From this observation, the VCO seemed to have a higher viscosity than the mixed VCO-PG, and PG. Furthermore, the presence of unabsorbed surfactants substantial fraction between the dispersed and water phases may lead to an extension of the particle size [45]. This may also describe why the emulsions prepared using mixed surfactants with HLB value of 10.2 exhibited the largest particle size diameter (Table 2). From the results, the particle size diameter of nanoemulsions loaded with mangostin extracts is governed by the HLB value of the surfactants and the type of the dispersed phase used.

The distribution width of nanoemulsions was revealed by the PDI (Table 2). PDI signifies the uniformity of droplet size within the formulation [23, 46]. PDI below 0.3 indicates good uniformity in the droplet size distribution after dilution with water [47]. PDI > 0.5 reveals a broad distribution [48]. In this study, the PDI values of formulations were found at a low range (0.17 to 0.37), indicating the narrow distribution of the droplet size within the formulation.

**Zeta potential.** Differences in the zeta potential of nanoemulsions formulated by different mixed surfactants varying in HLB values and different dispersive phases were found (Table 2). A dividing line between stable and unstable aqueous dispersions is generally taken at either ±30 mV. Particles with zeta potentials greater in negativity than -30 mV are

normally considered as stable [49]. From the results, the zeta potentials of all nanoemulsions varied from -39.4 mV to -58.1 mV, indicating the stable nanoemulsions gained. The largest negatively charged zeta potential was related to its strong repulsive forces, leading to a good dispersion system [39]. The negative surface charge of the nanoemulsion droplet might have resulted from the mixed Span 20/Tween 80 ions, forming hydrogen bonds between the mixed surfactants and water molecules in the boundary layer of the o/w emulsion. The largest negative zeta potential was obtained in all nanoemulsions containing mixed surfactants with an HLB value of 15.1 ($p < 0.05$). A slight lowering in the negative zeta potential was observed in nanoemulsions made from mixed surfactants with an HLB value of 10.2. It should be noted that nanoemulsions produced from the mixed emulsifiers with an HLB value of 12.6 had the lowest negative zeta potential ($p < 0.05$). This was in disagreement with the result of Ibrahim *et al.* [50], who reported that the zeta potential of the o/w emulsion prepared by blended fatty alcohol POE decreased with an increase in the HLB value. The difference might be due to the degree of compression of the electrical double layer around the emulsion droplets, which is influenced by the molecular structure and its distribution in the oil water interface. Typically, if surfactants can better form a hydrogen bond with water molecules or interact more truly with other molecules at the emulsion surface, a lower zeta potential value is obtained. Hence, the surface charge zeta potential results in the molecular behaviors at the oil droplet surface of emulsion. The type of oil dispersive phase also affected the zeta potential of nanoemulsions. A comparison at the same HLB value particularly at 10.2 showed that the nanoemulsions loaded with VCO-based mangostin extracts tended to have the highest negatively charged zeta potential, whereas the nanoemulsions loaded with PG-based mangostin extracts showed the lowest negative zeta potential (Table 2). This was probably due to the looser interaction between VCO and the mixed surfactants or other present molecules at the surface. The tightly packed nanoemulsion, led from more interaction of molecules at the interface with strong compression of the electrical double layer around the emulsion droplets when using PG-based mangostin extracts at the oil phase, was observed along with presenting the lower zeta potential and median particle size diameter of the nanoemulsions (Table 2).

## Nanoemulsion morphology

CLSM observation of nanoemulsions was done to confirm the results of light scattering data. CLSM image of coarse emulsions loaded with mangostin extracts prepared by a high speed homogenizer at 15,000 rpm for 5 min is depicted in Fig 1a, and nanoemulsions loaded with mangostin extracts emulsified by different oil phases and 5% mixed surfactants with varying HLB values is shown in Fig 1b–1j. The coarse emulsion was broad in distribution containing large globular droplets with an average micrometer in size, whereas the droplet sizes of all nanoemulsions were less than 100 nm (Fig 1). More uniform particle size distribution was also observed in nanoemulsions containing mixed emulsifiers with HLB values of 12.6 and 15.1. Droplets of nanoemulsions loaded with mangostin extracts made by mixed surfactants with HLB values ranging from 10.2–15.1 were globular and had a uniformly distributed nanoscale in size. A similar pattern was also found in nanoemulsions with different dispersed phases used. Pengon *et al.* [26] reported that the smallest droplet size of nanoemulsions made by a water compatible form of coconut oil through nanoemulsification using a high speed homogenizer was 162 nm. In this study, the smallest droplet size found in the nanoemulsions loaded with VCO-based mangostin extracts prepared under proper condition was 62 nm, which was lower than the previous report. Thus, this result indicated a suitable emulsion formulation and procedure to obtain small droplet sizes of less than 100 nm.

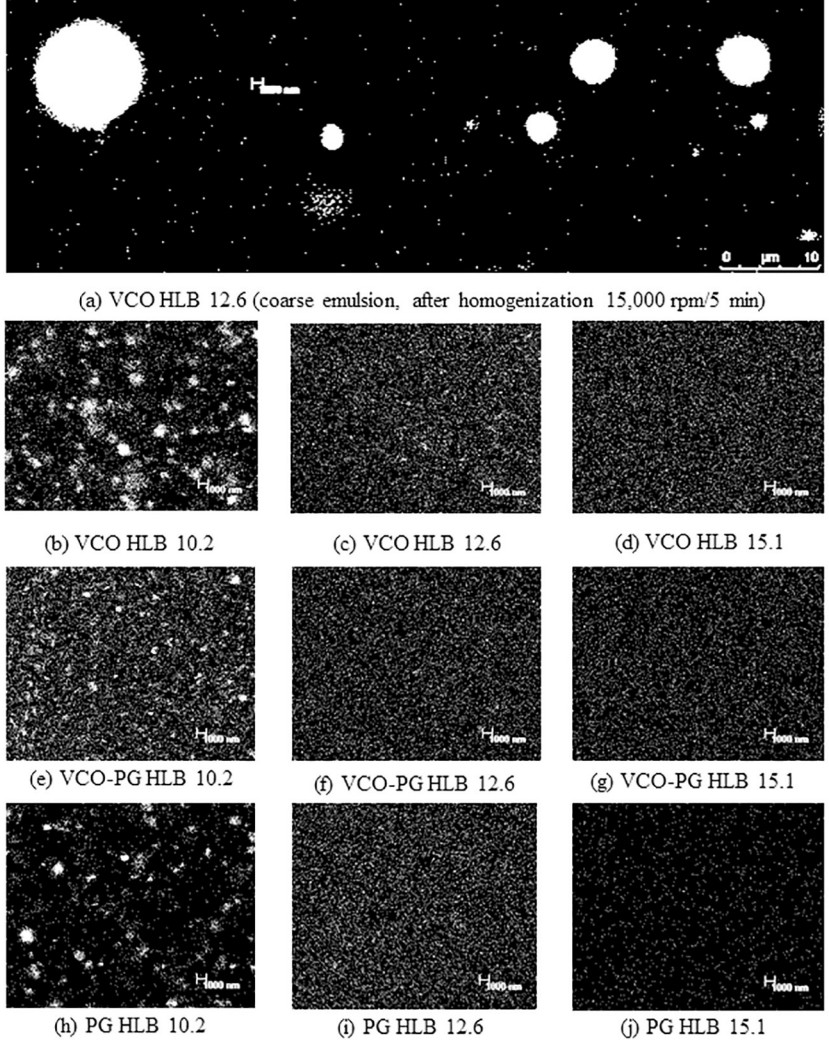

**Fig 1. Confocal laser scanning image of mangostin extract-loaded emulsions.** Coarse emulsions of VCO extract prepared using HLB 12.6 (a) Nanoemulsions of VCO extract prepared using HLB 10.2 (b), HLB 12.6 (c) and HLB 15.1 (d) Nanoemulsions of VCO-PG extract prepared using HLB 10.2 (e), HLB 12.6 (f) and HLB 15.1 (g) Nanoemulsions of PG extract prepared using HLB 10.2 (h), HLB 12.6 (i) and HLB 15.1 (j).

The higher the HLB value the greater the hydrophilicity of the surfactants, leading to a smaller droplet size [51, 52]. HLB provides important information on the promising formulation of nanoemulsions loaded with mangostin extracts. High HLB surfactants can form o/w nanoemulsions that progressively occupy the aqueous environment. It is of great interest to solubilize the hydrophobic active ingredient and drug for effective delivery, release, and skin penetration.

## Stability of nanoemulsions loaded with mangostin extracts

Physical stability during storage is of vital importance for the practical applications of nanoemulsion based delivery systems. All nanoemulsions were subjected to a freeze-thaw stability test, where the samples were subjected to extreme changes in storage temperature between -20 ˚C and 25 ˚C. The pH, particle size distribution, and zeta potential of each nanoemulsion were

measured before and after the freeze-thaw process. An increase in pH of all nanoemulsions was observed after the first freeze-thaw cycle, and then pH remained constant approximately at 4.5 afterwards as shown in Fig 2. This was caused by a two-phase formation, ice crystals and unfrozen solutes. The unfrozen solutes can be arranged into the remaining liquid volume at the end of the freeze process [53]. The concentration of unfrozen solutes may increase with the increasing freeze-thaw cycles and hence shifting the pH of the nanoemulsions. Nevertheless, the final pH after several freeze-thaw cycles was in the acceptable range of skin care (3.5–6) [54].

The changes in the zeta potentials of nanoemulsions formulated by different HLB mixed surfactants were also reported at the end of each freeze-thaw cycle (Fig 3). The reduction of negative zeta potential was observed in all nanoemulsions after the first freeze-thaw cycle. The negative zeta potential was continuously decreased with increasing freeze-thaw cycles. Mirhosseini *et al.* [55] pointed out that the zeta potential of the emulsion droplets was governed by

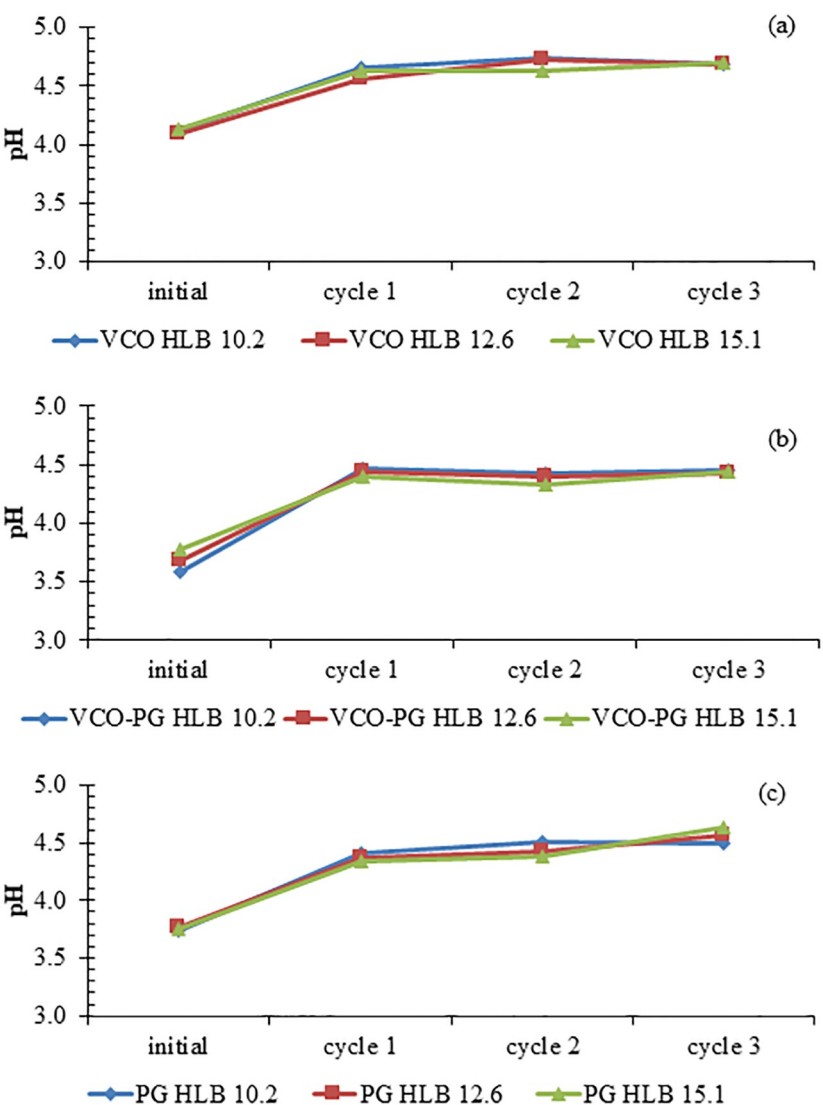

**Fig 2. Effect of freeze-thaw cycle on pH stability of mangostin extract-loaded nanoemulsions.** VCO extract (a), VCO-PG extract (b), and PG extract (c). Bars represent the standard deviation from triplicate determinations.

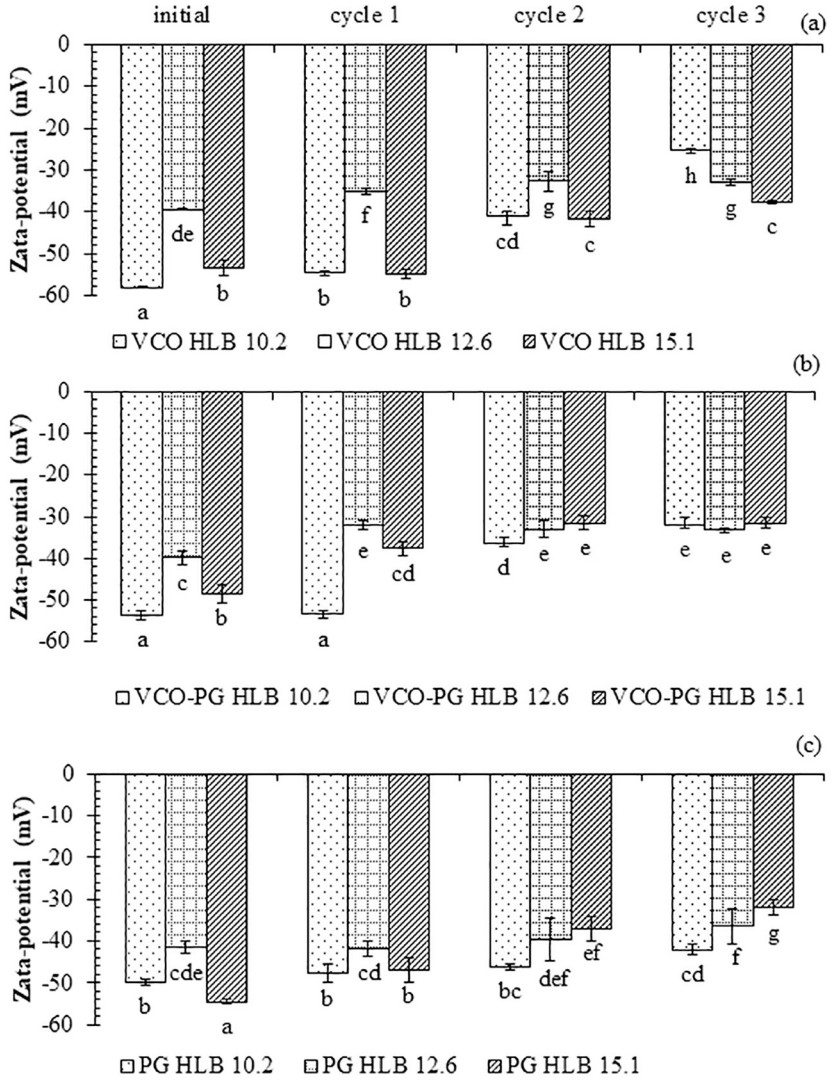

**Fig 3. Effect of freeze-thaw cycle on zeta potential stability of mangostin extract-loaded nanoemulsions.** VCO extract (a), VCO-PG extract (b), and PG extract (c). Bars represent the standard deviation from triplicate determinations. Different letters on the bars indicate significant differences ($p < 0.05$).

the final pH. Therefore, lowering zeta potentials after being subjected to freeze-thaw might be because of pH changes. In addition, the molecular movement at the oil-water interface of surfactants during temperature fluctuations might induce the reduction in negative zeta potential. However, the final zeta potential of nanoemulsions after multiple freeze-thaw cycles was still within the acceptable range ($< -30$ mV) and normally considered as stable. There were no signs of phase separation for PG and mixed VCO-PG- based nanoemulsions, indicating stability even under severe conditions. During the freeze-thaw process, small ice crystals can be generated and became larger after several freeze-thaw cycles.

The droplet size distributions of nanoemulsions loaded with mangostin extracts using VCO, mixed VCO-PG, and PG-based extracts at dispersive phases formulated by different HLB values of surfactants are shown in Fig 4. Nanoemulsions made by mixed surfactants with an HLB value of 15.1 showed the smallest particle size diameter, and the size increased

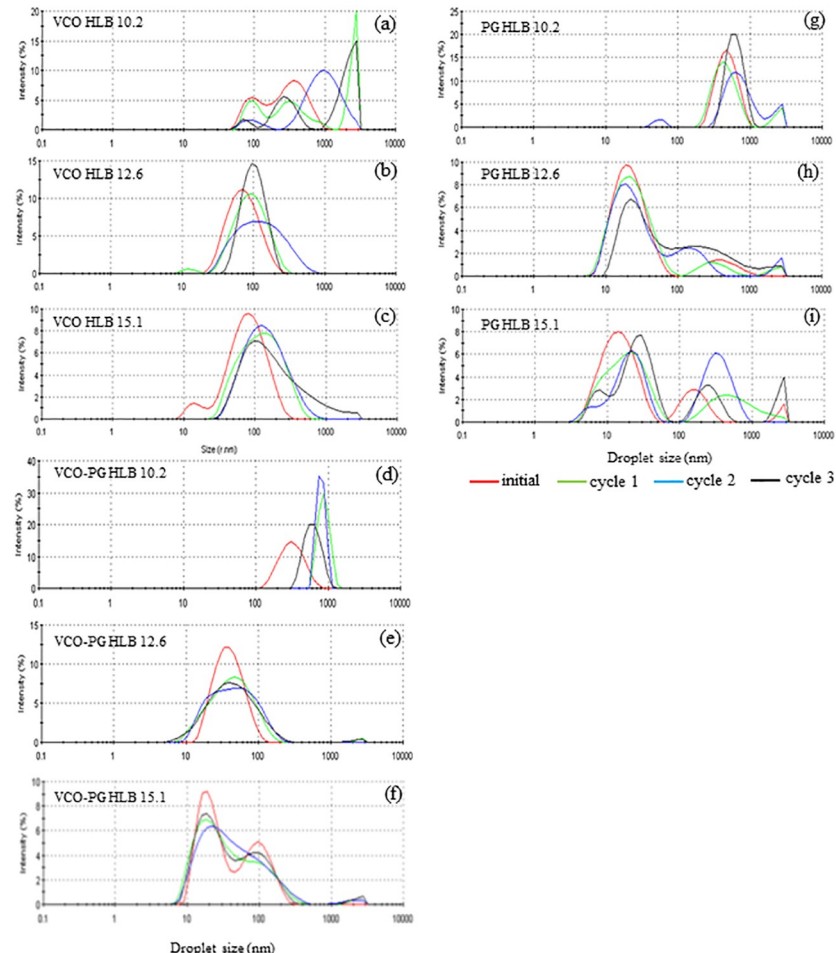

**Fig 4. Effect of freeze-thaw cycle on particle size distribution of mangostin extract-loaded nanoemulsions.** VCO extract: HLB 10.2 (a), HLB 12.6 (b), and HLB 15.1 (c) VCO-PG extract: HLB 10.2 (d), HLB 12.6 (e), and HLB 15.1 (f) PG extract: HLB 10.2 (g), HLB 12.6 (h), and HLB 15.1 (i).

negligibly after several freeze-thaw cycles (Fig 5). The obvious changes in particle size diameter and its distribution were detected in nanoemulsions loaded with VCO-based and mixed VCO-PG-based mangostin extracts produced by mixed surfactants having an HLB value of 10.2 (Figs 4 and 5). This corresponded well with the initial median particle size and zeta potential (Table 2), where such systems showed the largest particle size diameter and broad particle size distribution. Ariyaprakai and Tananuwong [56] reported that the median particle size diameter of coconut oil and corn oil nanoemulsions stabilized by various types of sucrose esters and Tweens remained constant after one freeze-thaw cycle. However, the particle size distribution of coconut oil and corn oil nanoemulsions was slightly shifted after freezing and thawing. The different results could have originated from the different number of freeze-thaw cycles employed. Destabilization of nanoemulsions is typically caused by the Ostwald ripening, flocculation, and coalescence [25]. The interfacial layers of the small molecular surfactants used in this study was unable to provide enough mechanical strength to prevent coalescence in the nanoemulsions loaded with mangostin extracts after several freeze-thaw cycles. It should be noted that the droplet growth was dependent on the HLB value of the surfactants. The

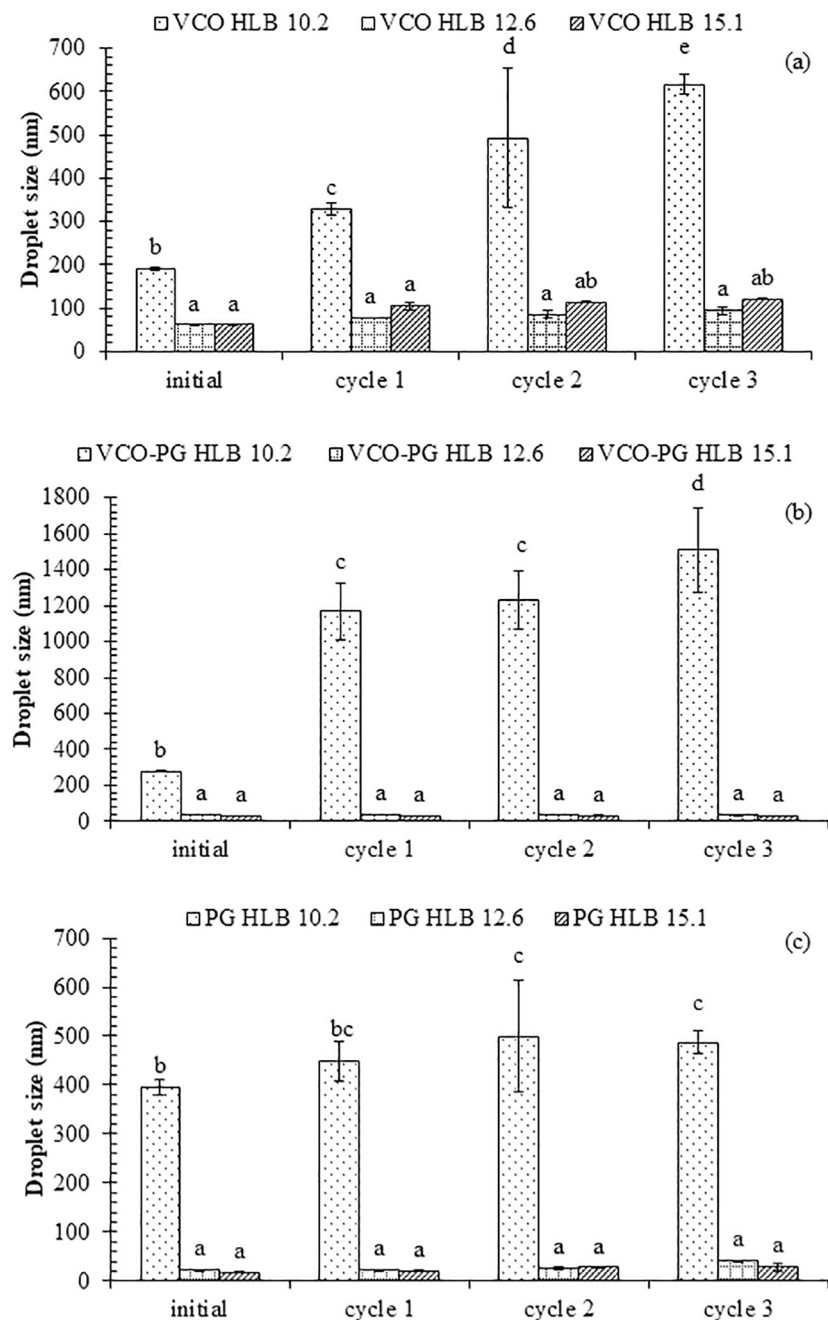

**Fig 5. Effect of freeze-thaw cycle on mean particle size of mangostin extract-loaded nanoemulsions.** VCO extract (a), VCO-PG extract (b), and PG extract (c). Bars represent the standard deviation from triplicate determinations. Different letters on the bars indicate significant differences ($p < 0.05$).

droplet size slightly increased with increasing freeze-thaw cycles and the growth rate decreased with increasing HLB value of the surfactant. Therefore, the emulsion stability can be improved by stabilization with a proper HLB emulsifier. From this study, nanoemulsions loaded with mangostin extracts containing mixed emulsifiers having an HLB value of 15.1 was considered as the most promising shelf stable formulation.

**Table 3. Antioxidant activity of mangostin bulk extracts recovered by virgin coconut oil (VCO), mixed VCO-prolylene glycol (PG) and PG compared to their corresponding nanoemulsions.**

| Antioxidant activity | Extractant | Bulk extract | Nanoemulsion* |
|---|---|---|---|
| DPPH assay[1] ($EC_{50}$, mg/mL) | VCO | $5.64^c \pm 0.22$ | $4.27^c \pm 0.17$ |
| | Mixed VCO-PG | $1.65^a \pm 0.07$ | $1.30^a \pm 0.17$ |
| | PG | $2.38^b \pm 0.07$ | $1.94^b \pm 0.25$ |
| $H_2O_2$ assay[2] ($EC_{50}$, mg/mL) | VCO | $6.65^c \pm 0.33$ | $4.44^c \pm 0.35$ |
| | Mixed VCO-PG | $4.15^a \pm 0.19$ | $2.66^a \pm 0.17$ |
| | PG | $4.62^b \pm 0.08$ | $3.61^b \pm 0.09$ |
| Reducing power (mg AAE /100 g) | VCO | $61.56^a \pm 10.72$ | $86.60^a \pm 3.82$ |
| | Mixed VCO-PG | $279.22^c \pm 11.41$ | $348.92^c \pm 7.35$ |
| | PG | $220.79^b \pm 23.49$ | $281.40^b \pm 5.76$ |

Values are given as mean±standard deviation from triplicate determinations.

Different letters in the same column indicate significant differences ($p < 0.05$).

[1]$EC_{50}$ value of DPPH•scavenging activity of α-tocopherol was 22.86 μg/mL.

[2]$EC_{50}$ value of $H_2O_2$ scavenging activity of ascorbic acid was 162.2 μg/mL.

AAE = ascorbic acid equivalents

*Emulsions were prepared using 10% (w/w) mixed emulsifiers (Span 20 and Tween 20; HLB = 15.1) and 10% crude mangostin extracts using a high-speed homogenizer (15,000 rpm) for 5 min and an ultrasonic homogenization (20 kHz, 750 Watts, 70% amplitude) for 10 min.

### Antioxidant activities of nanoemulsions loaded with mangostin extracts

The antioxidative activities, including DPPH• scavenging activity, $H_2O_2$ scavenging activity and reducing power, of bulk mangostin extracts and their corresponding nanoemulsions produced by mixed emulsifiers with an HLB value of 15.1 are presented in Table 3. For DPPH• and $H_2O_2$ scavenging activities, the $EC_{50}$ values were reported. Both bulk mangostin extracts and their nanoemulsions had inferior activities to positive controls, α-tocopherol for DPPH assay ($EC_{50}$ = 22.86 μg/mL) and ascorbic acid for $H_2O_2$ assay ($EC_{50}$ = 162.2 μg/mL). All nanoemulsions provided greater DPPH• and $H_2O_2$ scavenging activities than their bulk extracts ($p < 0.05$). The DPPH• scavenging activities of VCO-, mixed VCO-PG-, and PG-based nanoemulsions were higher than their correspondence bulk extracts for 1.32, 1.27, and 1.22-fold, respectively. For $H_2O_2$ scavenging activities, the values were 1.50, 1.56, and 1.28 times higher than the bulk extracts, respectively. Aungst [57] mentioned that small droplet size increases the specific surface area and can improve bioactive release, transparency, and bioavailability. It was found that both mixed VCO-PG bulk extracts and its nanoemulsions showed superior DPPH• and $H_2O_2$ scavenging activities when compared to the others within the same system (bulk extracts or nanoemulsions). For bulk extracts, the DPPH• scavenging activity of mixed VCO-PG-based extracts was higher than the VCO-based- and PG-based extracts by 3.96 and 1.44 times, respectively (Table 2). In addition, the $H_2O_2$ scavenging activity of mixed VCO-PG extracts was greater than the VCO-based and PG-based extracts by 1.60 and 1.11-fold, respectively. This was probably due to the effect of extractants, which can recover different types of phenolic compounds with different radical scavenging activities [58]. Among nanoemulsions, the highest DPPH• and $H_2O_2$ scavenging activities were found in the mixed VCO-PG-based nanoemulsions ($p < 0.05$). VCO-loaded nanoemulsions rendered the lowest scavenging activities against both DPPH• and $H_2O_2$ ($p < 0.05$). The difference in DPPH• and $H_2O_2$ scavenging activities among nanoemulsions made by three dispersive phases was due to the particle size of the nanoemulsions in which the larger particle size was obtained from all VCO-based nanoemulsions. Surprisingly, the smallest particle size of PG nanoemulsions showed lower DPPH•

and $H_2O_2$ scavenging activities (Table 3). The smaller droplets of nanoemulsions with larger specific surface area might be more prone to oxidation than the larger droplets [59]. This phenomenon may result in a reduction of overall antioxidative activity of the system with tiny droplets. Anh-Ha *et al*. [60] found that the DPPH$^\bullet$ scavenging efficiency of lycopene nanoemulsions increased with the decrease in droplet size up to 100 nm. Nguyen *et al*. [61] reported that the smaller the droplet size the greater the antioxidant activity. From the results, the mixed VCO-PG-loaded nanoemulsions with moderate particle size diameters scavenged more DPPH$^\bullet$ and $H_2O_2$. Therefore, this can be used as a promising delivery system for bioactive polyphenol in food supplements, pharmaceuticals and cosmetics.

The same trend was found in the reducing power, where nanoemulsions had a higher activity than their corresponding bulk extracts (Table 3). The VCO-, mixed VCO-PG-, and PG-based nanoemulsions exhibited higher reducing ability than their corresponding bulk extracts by 1.40, 1.25, and 1.27 times, respectively. This could be explained by the principle of droplet size similar to the DPPH$^\bullet$ and $H_2O_2$ scavenging results. Among bulk extracts, the highest reducing capacity was also found in mixed VCO-PG extracts ($p < 0.05$) at 4.56 and 1.26 times greater than the VCO and PG extracts, respectively. The highest reducing power was found in nanoemulsions made with mixed VCO-PG extracts, followed by PG and VCO extracts (Table 3). This was probably due to the smaller size of the nanoemulsion droplet when it was prepared by mixed VCO-PG. Also, the different types of phenolics recovered by different extractants can be found among samples [58]. Our previous study found that the blending of 40–70% PG and VCO can improve the extraction of α-mangostin and γ-mangostin from mangosteen peels resulting in the improvement of antioxidant activities [5].

## Antibacterial activities

The inhibitory effects of bulk mangostin extracts and nanoemulsions loaded with mangostin extracts against gram-negative bacteria (*E. coli*) and gram-positive bacteria (*S. aureus*) were reported by a minimum inhibitory concentration (MIC) (Table 4). The nanoemulsions showed a higher inhibition against both organisms than their corresponding bulk extracts. The *E. coli* inhibition of VCO-, mixed VCO-PG-, and PG-extract nanoemulsions was 2.0, 1.97, and 1.97 times higher than the bulk extracts, respectively. For *S. aureus* inhibition, VCO-, mixed VCO-PG-, and PG-extract nanoemulsion showed 1.97, 1.97, and 2.0 times higher than

**Table 4. Antibacterial activity of mangostin bulk extracts recovered by virgin coconut oil (VCO), mixed VCO-prolylene glycol (PG) and PG compared to their corresponding nanoemulsions.**

| Antibacterial activity | Extractant | MIC values (mg/mL) | |
|---|---|---|---|
| | | **Bulk extract** | **Nanoemulsions*** |
| *E. coli*[1] | VCO | 3.13 | 1.56 |
| | Mixed VCO-PG | 1.56 | 0.79 |
| | PG | 1.56 | 0.79 |
| *S. aureus*[2] | VCO | 1.56 | 0.79 |
| | Mixed VCO-PG | 1.56 | 0.79 |
| | PG | 3.13 | 1.56 |

Values are given as mean from triplicate determinations.

[1]MIC value against *E. coli* of ampicillin was 4 µg/mL.

[2]MIC value against *S. aureus* of ampicillin was 4 µg/mL.

*Emulsions were prepared using 10% (w/w) mixed emulsifiers (Span 20 and Tween 20; HLB = 15.1) and 10% crude mangostin extracts using a high-speed homogenizer (15,000 rpm) for 5 min and an ultrasonic homogenization (20 kHz, 750 Watts, 70% amplitude) for 10 min.

the bulk extracts, respectively. The improved antimicrobial activity of nanoemulsions loaded with mangostin extracts may be due to the increased mangostin level in the aqueous phase [15]. This finding was in agreement with Topuz *et al.* [38], who reported on the better antimicrobial activity of anise oil nanoemulsions than the bulk anise oil. However, some contradictory results have been reported. Gomes *et al.* [62] reported that the inhibiting effects of essential oil nanoemulsions against *Salmonella* spp. and *Listeria* spp. were comparable with the bulk essential oil. This study confirmed that nanoemulsions can be used as an effective polyphenol-based delivery system with antimicrobial activity.

Among bulk extracts, both mixed VCO-PG and PG extracts showed the greatest *E. coli* inhibitory efficiency (MIC = 1.56 mg/mL), whereas the lowest such activity was found in the VCO extract (MIC = 3.13 mg/mL). In contrast, the highest inhibitory efficiency against *S. aureus* was found in the VCO and mixed VCO-PG extracts (MIC = 1.56 mg/mL) and the lowest was found in the PG extract (MIC = 3.13 mg/mL) (Table 4). This similar trend was found in nanoemulsions made by their corresponding bulk extracts. It should be noted that both mixed VCO-PG extract and its nanoemulsions showed the best inhibitory effect toward both gram-negative bacteria, *E. coli*, and gram-positive bacteria, *S. aureus*, indicating the powerful antimicrobial activity of this extract. Interestingly, the lowering in the *S. aureus* inhibitory effect of both PG bulk extract and its nanoemulsions were observed when compared to the VCO- and mixed PG-based extracts and their nanoemulsions, which was opposite to *E. coli*. This was probably due to the greater effect of bioactive compounds found originally in VCO on *S. aureus* than on mangosteen polyphenols and smaller particle sizes. VCO contains natural monolaurin and lauric acid behaving with a synergistic inhibitory effect on food borne pathogens [63]. VCO-based nanoemulsions as a delivery system may diffuse more deeply throughout the thick peptidoglycan cell membrane of gram-positive bacteria than other delivery systems that have been tested. As a result, the superior *S. aureus* inactivation was noticed in this VCO-based nanoemulsion system. Therefore, the antimicrobial effects of mangosteen pericarp extracts were varied depending on the dispersive phase (selective polyphenol recovery), the delivery system (bulk extracts or nanoemulsions), and the microorganism species.

## Properties of emulgels and nanoemulgels

Emulgels and nanoemulgels can be widely applied in food and cosmetic formulations [64]. The emulgels and nanoemulgels of mangostin extracts recovered by VCO, mixed VCO-PG and PG were evaluated for their clarity, glitter, homogeneity, pH, color, and viscosity (Table 5). Emulgels and nanoemulgel prepared in this study can be used as a guidance to potentially investigate the utilization of such extracts as a functional ingredient in skin care-based products. PG-based extracts of both emulgels and nanoemulgels showed superior clarity, glitter, homogeneity, color, and viscosity when compared to other formulations. The PG-based nanoemulgels also showed a clear yellow viscous gel with a glittery and homogeneous appearance. The mixed VCO-PG formulation was a clear yellow-white viscous gel with a smooth and homogeneous appearance, while the VCO formulation was a creamy yellow-white viscous preparation with a smooth and homogeneous appearance. Negligible differences among emulgels and nanoemulgels prepared by various extracts were observed. The pH values of all prepared gels ranged from 5.66 to 5.95, which are considered as safe to avoid the risk of irritation upon application to the skin because normal adult skin pH is around 4–6 [54]. The highest viscosity of the emulgels and nanoemulgels prepared by the VCO-based extract was found (p < 0.05) (12,588 and 12,093 cps, respectively). A higher viscosity of the emulgels and nanoemulgels when compared to the liquid emulsion system can increase the retention time of active components on the skin surface and promote better absorption [65]. No significant

**Table 5. Appearance and characteristics of emulgels and nanoemulgels loaded with mangostin extracts.**

| Parameter | Emulgel | | | Nanoemulgel | | |
|---|---|---|---|---|---|---|
| | VCO | Mixed VCO-PG | PG | VCO | Mixed VCO-PG | PG |
| Clarity | + | ++ | +++ | + | ++ | +++ |
| Glitter | Non | + | +++ | Non | + | +++ |
| Homogeneity | Good | Good | Good | Good | Good | Good |
| pH | 5.74 ± 0.04 [bc] | 5.66 ± 0.07 [ab] | 5.57 ± 0.12 [a] | 5.88 ± 0.06 [cd] | 5.82 ± 0.10 [cd] | 5.95 ± 0.07 [d] |
| Viscosity (cps) | 12588 ± 291 [b] | 10676 ± 812 [a] | 10003 ± 655 [a] | 12093 ± 754 [b] | 10423 ± 901 [a] | 9958 ± 556 [a] |
| $L^*$ | 53.00 ± 0.93 [d] | 34.99 ± 0.50 [b] | 5.26 ± 0.22 [a] | 52.51 ± 0.80 [d] | 47.31 ± 0.49 [c] | 27.42 ± 0.31 [a] |
| $a^*$ | 8.55 ± 0.47 [a] | -4.75 ± 0.21 [c] | 2.81 ± 0.19 [d] | -8.47 ± 0.34 [a] | -7.66 ± 0.23 [b] | -3.26 ± 0.11 [d] |
| $b^*$ | 35.59 ± 1.99 [d] | 24.96 ± 0.33 [c] | 14.59 ± 0.39 [a] | 34.03 ± 1.34 [d] | 33.75 ± 0.03 [d] | 16.91 ± 1.00 [b] |

Values are given as mean±standard deviation from triplicate determinations.

Different letters in the same row indicate significant differences ($p < 0.05$).

*Mangostin extracts were recovered by virgin coconut oil (VCO), mixed VCO-propylene glycol (PG) and PG.

For clarity and glitter, + = medium, ++ = high, and +++ = very high.

difference in gel viscosity was detected in the mixed VCO-PG-based- and PG-based emulgels and nanoemulgels when compared with the same system ($p > 0.05$). Regarding the same extract used, nanoemulgels had lower viscosity than emulgels, indicating the influence of the droplet size on the viscosity of the resulting gel. It should be taken into account that viscosity of the PG and mixed VCO-PG extract-based nanoemulgels in this study were similar to Keto-profen emulgels (9326–9756 cps) [30]. This viscosity was suitable for cosmetic formulation for topical use. In both the emulgels and nanoemulgels, gels made with the VCO extract exhibited the highest $L^*$, $a^*$, and $b^*$ values followed by gels made with the mixed VCO-PG and PG extract (Table 5). Slight differences in the $L^*$ and $b^*$ values were observed among the emulgels and nanoemulsgels, whereas a marked change in the $a^*$ value was observed between individual emulgels and nanoemulgels made from the same extract. These indicated the impact of both the dispersed phase and its particle size on the color of the gel. From the results, preparation of nanoemulgels is of great interest because of its effect on optimal gel appearance, viscosity, and color.

**In vitro mangostin release study of emulgels and nanoemulsgels.** *In vitro* mangostin release studies were carried out for all the formulations. Table 6 shows the mangostin release

**Table 6. Mangostin release from emulgel and nanoemulgel.**

| Formulation | Extractant* | Mangostins release (%) | |
|---|---|---|---|
| | | α-mangostin | γ-mangostin |
| Emulgel | VCO | 77.14 ± 10.02[a] | 83.27 ± 9.42[a] |
| | Mixed VCO-PG | 73.78 ± 3.81[a] | 77.96 ± 3.31[a] |
| | PG | 89.20 ± 4.82[b] | 88.84 ± 4.44[ab] |
| Nanoemulgel | VCO | 89.26 ± 5.95[b] | 88.38 ± 7.96[ab] |
| | Mixed VCO-PG | 92.17 ± 4.56[b] | 86.94 ± 5.01[ab] |
| | PG | 94.19 ± 5.40[b] | 92.84 ± 8.66[b] |

Values are given as mean±standard deviation from triplicate determinations.

Different letters in the same column indicate significant differences ($p < 0.05$).

*Mangostin bulk extract was recovered by virgin coconut oil (VCO), mixed VCO-propylene glycol (PG) and PG.

from emulgels and nanoemulgels. With the same extract used, the nanoemulgels were able to release α-mangostin and γ-mangostin from the matrix better than the emulgels ($p < 0.05$), indicating an improved controlled release of the active compound from this vehicle. Reeves *et al.* [66] indicated that curcumin-loaded nanogel formulation was 70–85% more effective in inhibiting cancer cell growth, at concentrations lower than $IC_{50}$ of free curcumin. From the results, the releasing rate of mangosteen polyphenols was governed by both the bioactive dispersed media and its particle size in the gel matrix. The release of both mangostins from its emulsified emulgel formulation was ranked in the following descending order: PG extract > VCO extract ≈ mixed VCO-PG extract. A similar trend was also found in emulsified nanoemulgels (Table 6). Although PG is defined as "generally regarded as safe", in reality, its toxicity is still questioned [67]. As a consequence, the lowered PG content in the extract by mixing with VCO is a promising strategy to reduce its content in cosmetics which will be safer for humans. Furthermore, VCO naturally contains monolaurin and lauric acid, which can synergize the antibacterial effect of mangosteen polyphenols (Table 4). Thus, the nanoemulgels containing mangostin extracts recovered by mixed VCO-PG was considered as the superior formulation to all gels. The advantages of a delivery system through nanoemulgels include good adhesion to the surface of the skin and high solubilizing capacity leading to a larger concentration gradient towards the skin. Nanoemulgels can release oily droplets containing mangosteen polyphenols from the gel matrix upon skin contact and subsequently penetrate the skin without a transfer via the hydrophilic phase of nanoemulsions [68–69].

## Conclusions

Nanoemulsions loaded with mangostin extracts prepared from mangosteen peel extracts recovered by VCO, mixed VCO-PG, and PG at the dispersed phase containing mixed surfactants (Tween20/Span20) having an HLB value of 15.1 were successfully produced by ultrasonication. The obtained nanoemulsions were globular and uniformly distributed in nanoscale, with the average droplet size less than 100 nm. The zeta potentials of the particles exerted the largest negative charge, which was considered as a stable dispersion. All nanoemulsions made by mixed surfactant with an HLB value of 15.1 were stable against multiple freeze-thaw cycles. Furthermore, the smaller droplet sizes of the nanoemulsions exhibited superior antioxidant and antibacterial activities when compared to their corresponding bulk extracts. Thus, nanoemulsions loaded with mangostin extracts can be used as a promising delivery system for bioactive polyphenol in food supplements, pharmaceuticals, and cosmetics. A prospective nanoemulgel using nanoemulsions loaded with mangostin extracts and oligosaccharide as a gelling agent was developed. The nanoemulgel showed a greater increase in *in vitro* mangostin release than its emulgel. Considering its gel appearance, viscosity, and mangostin release, the mixed VCO-PG nanoemulgel was classified as an excellent formulation for cosmetic-based gel application.

## Author Contributions

**Conceptualization:** Worawan Panpipat, Manat Chaijan.

**Data curation:** Chatchai Sungpud.

**Writing – original draft:** Chatchai Sungpud, Worawan Panpipat, Manat Chaijan, Attawadee Sae Yoon.

**Writing – review & editing:** Chatchai Sungpud, Worawan Panpipat, Manat Chaijan, Attawadee Sae Yoon.

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
