## [Decision Letter · Decision Letter 0]

12 Dec 2019

PONE-D-19-30992

Techno-Biofunctionality of Mangostins Loaded Virgin Coconut Oil Nanoemulsion and Nanoemulgel

PLOS ONE

Dear Dr. Panpipat,

Thank you for submitting your manuscript to PLOS ONE. After careful consideration, we feel that it has merit but does not fully meet PLOS ONE’s publication criteria as it currently stands. Therefore, we invite you to submit a revised version of the manuscript that addresses the points raised during the review process.

We would appreciate receiving your revised manuscript by Jan 26 2020 11:59PM. To enhance the reproducibility of your results, we recommend that if applicable you deposit your laboratory protocols in protocols.io, where a protocol can be assigned its own identifier (DOI) such that it can be cited independently in the future. For instructions see: http://journals.plos.org/plosone/s/submission-guidelines#loc-laboratory-protocols

We look forward to receiving your revised manuscript.

Kind regards,

Walid Elfalleh, Ph.D

Academic Editor

PLOS ONE

Journal Requirements:

1. We noticed you have some minor occurrence of overlapping text with the following previous publication(s), which needs to be addressed:

https://www.sciencedirect.com/science/article/pii/S0023643819309612?via%3Dihub

https://www.sciencedirect.com/science/article/pii/B9780123741240000247

In your revision ensure you cite all your sources (including your own works), and quote or rephrase any duplicated text outside the methods section. Further consideration is dependent on these concerns being addressed.

Reviewers' comments:

Reviewer's Responses to Questions

**Comments to the Author**

1. Is the manuscript technically sound, and do the data support the conclusions?

Reviewer #1: Yes

Reviewer #2: Yes

Reviewer #3: Yes

Reviewer #4: Yes

2. Has the statistical analysis been performed appropriately and rigorously? 

Reviewer #1: Yes

Reviewer #2: Yes

Reviewer #3: Yes

Reviewer #4: Yes

3. Have the authors made all data underlying the findings in their manuscript fully available?

Reviewer #1: Yes

Reviewer #2: Yes

Reviewer #3: Yes

Reviewer #4: Yes

4. Is the manuscript presented in an intelligible fashion and written in standard English?

Reviewer #1: Yes

Reviewer #2: Yes

Reviewer #3: Yes

Reviewer #4: Yes

5. Review Comments to the Author

Reviewer #1: This manuscript studied techno-biofunctional characteristics of mangostins loaded nanoemulsion and (nano)emulgel. The study is interesting. This manuscript is suitable for publication in PLOS one with major revision.

Comments:

1. The title of the paper should be modified since the work is devoted to mangostin extracts but not to mangostins.

2. The authors should add the contents of α-mangostin and γ-mangostin in nanoemulsion and (nano)emulgel.

3. In the determination of antioxidant activities, The α- tocopherol was not only the positive control of DPPH● scavenging activity, but also the positive control of other antioxidant activities.

4. In the antioxidant activities and antibacterial activity, Blank control of nanoemulsion without mangostin extracts should be added.

5. The authors should add and explain the EC50 of positive control and blank control.

6. There are so many language errors in this manuscript. We recommend the authors asking a professional editor or native English speaker to fully improve the language of this manuscript.

Reviewer #2: The manuscript mainly presents a Techno-Biofunctionality of Mangostins Loaded Virgin Coconut Oil Nanoemulsion and Nanoemulgel. I have read this paper with great interest as I can see the value of the approach investigated by the authors. However, I have a few issues in some aspects:

The manuscript has a clear problem statement and solution, but it needs some revision for the English language.

In fig.3 and fig.5, please indicate significant differences by different letters (statistical analysis).

Could you explain more please results given in table 1.

I would advise authors to complete the conclusion by adding the utility of this work.

Reviewer #3: In this manuscript named “Techno-Biofunctionality of Mangostins Loaded Virgin Coconut Oil Nanoemulsion and Nanoemulgel” Chatchai Sungpud et al are presenting a detailed technical and biofunctional comparison between Mangostins loaded nanoemulsions prepared from mangosteen peel extracts recovered by 3 different extractants (VCO, PG, and mixed VCO-PG). Based on an extensive characterization, the authors established that Mangostins loaded nanoemulsions prepared from mangosteen peel extracts recovered by mixed VCO-PG nanoemulsions were globular and uniformly distributed in nanoscale with the average droplet of less than 100 nm with negatively charged Zeta potentials of particles. The authors also tested the antioxidant and antibacterial activities of the nanoemulsions.

They also described the gel appearance, viscosity and mangostins release of mixed VCO-PG nanoemulgels as an excellent formulation for cosmetic based gel application.

Even though the manuscript is well written, nicely detailed and organized, I have few remarks:

*Abstract:

Modifications should be made to showcase the importance of your work and the potential of further interesting applications.

Less technical details and numbers should be included in the abstract.

Changes in the structure of the sentences should be introduced to make the abstract easier to follow.

*Material and Methods:

Nicely detailed techniques, explanations, and equations.

Since table 4 is the first table you refer to it, it should be named as table 1.  

*Results and discussion:

In “Stability of mangostins loaded nanoemulsions” paragraph, line 367, the authors described that freeze-thaw cycles affected the pH of the nanoemulsion and confirm that the final pH is still in the acceptable range of skincare (line 369), while the original pH is out of this range (less than 4).

In “Antibacterial activities” paragraph, line 447, negative controls containing the extractant should be tested side by side with the corresponding nanoemulsions.

Figure 1: The illustration of the panel with the same magnification is very important but it is very hard to see the nanoemulsion of PG extract prepared using HLB 15.1 (j) at this magnification. An additional picture with higher magnification should be added to proof that the nanoemulsions are present but in a smaller size.

A number of typos through the manuscript should be corrected.

*Finally, this manuscript described an efficient strategy to prepare Mangostins loaded nanoemulsions prepared from mangosteen peel extracts with moderate particle size diameter scavenged more DPPH and H2O2 which can be used as a promising delivery system for bioactive polyphenol in food supplement, pharmaceutical and cosmetic. Furthermore, the VCO-PG nanoemulgel was classified as an excellent formulation for cosmetic based gel application.

Reviewer #4: Overall, this is a clear, concise, and well-written manuscript. The introduction is relevant and theory based. Sufficient information about the previous study findings is presented for readers to follow the present study rationale and procedures. The methods are generally appropriate, and well described. Overall, the results are clear and convincing. Overall, this is a high quality manuscript that has implications for the theoretical basis, devet.

Tables are cleae as well as figures.

This work is concidered for publication.

6. PLOS authors have the option to publish the peer review history of their article (what does this mean?). If published, this will include your full peer review and any attached files.

Reviewer #1: No

Reviewer #2: No

Reviewer #3: No

Reviewer #4: No

---

## [Author Response · Author response to Decision Letter 0]

18 Dec 2019

Response to Reviewers 

All points raised by the academic editor and reviewers were carefully addressed and answered point-by-point. A revision was made in highlighted red fonts. The revised manuscript was carefully proofread for English by a native speaker, Mr. Dave Chang from USA. The revised manuscript was carefully prepared to meet PLOS ONE's style requirements.

Journal Requirements:

Ans: The revised manuscript was carefully prepared to meet PLOS ONE's style requirements.

1. We noticed you have some minor occurrence of overlapping text with the following previous publication(s), which needs to be addressed:

https://www.sciencedirect.com/science/article/pii/S0023643819309612?via%3Dihub

https://www.sciencedirect.com/science/article/pii/B9780123741240000247

In your revision ensure you cite all your sources (including your own works), and quote or rephrase any duplicated text outside the methods section. Further consideration is dependent on these concerns being addressed.

Ans: The overlapping text with the article (https://www.sciencedirect.com/science/article/pii/S0023643819309612?via%3Dihub) was rewritten and this reference was added instead. [64. Chang WC, Hu YT, Huang Q, Hsieh SC, Ting Y. Development of a topical applied functional food formulation: Adlay bran oil nanoemulgel. LWT. 2020;117:108619.]

The reference from https://www.sciencedirect.com/science/article/pii/B9780123741240000247) was added. [25. Weiss J, Gaysinsky S, Davidson M, McClements J. Nanostructured encapsulation systems: food antimicrobials. In Global issues in food science and technology 2009 Jan 1 (pp. 425-479). Academic Press.]

All our own sources were cited. 

Reviewer #1: This manuscript studied techno-biofunctional characteristics of mangostins loaded nanoemulsion and (nano)emulgel. The study is interesting. This manuscript is suitable for publication in PLOS one with major revision.

Comments:

1. The title of the paper should be modified since the work is devoted to mangostin extracts but not to mangostins.

Ans: The title was changed to “Techno-biofunctionality of mangostin extract-loaded virgin coconut oil nanoemulsion and nanoemulgel”. Also, the term “mangostin extracts” were used throughout the text.

2. The authors should add the contents of α-mangostin and γ-mangostin in nanoemulsion and (nano)emulgel.

Ans: The contents of �-mangostin and γ-mangostin in nanoemulsion and (nano)emulgel were added in P6 and P11, respectively.

3. In the determination of antioxidant activities, The α- tocopherol was not only the positive control of DPPH● scavenging activity, but also the positive control of other antioxidant activities.

Ans: The α- tocopherol can be used as the positive control for all antitoxidant activities. However, in this study, the α- tocopherol was used as the positive control of DPPH● scavenging activity due to its excellent free radical scavenging activity. Ascorbic acid was used as the positive control for hydrogen peroxide scavenging activity and reducing power due to the its ability to scavenge hydrogen peroxide and its reducing property.

4. In the antioxidant activities and antibacterial activity, Blank control of nanoemulsion without mangostin extracts should be added.

Ans: We have originally stated in the methods about the blank control. All tests had the blank control without the sample. Unfortunately, the blank controls of nanoemulsion without mangostin extracts were not prepared. We think that the nanoemulsion was prepared using the same components and contents. So, the different results should come from the compounds in the extracts only. Also, the activities of bulk extracts were also compared. 

5. The authors should add and explain the EC50 of positive control and blank control.

Ans: The EC50 values of positive control for DPPH assay and hydrogen peroxide assay were originally presented in the footnote of Table 2 as “1EC50 value of DPPH●scavenging activity of �-tocopherol was 22.86 �g/mL. 2EC50 value of H2O2 scavenging activity of ascorbic acid was 162.2 �g/mL.” We also added in the discussion that “Both bulk mangostin extracts and their nanoemulsions had inferior activities to positive controls, �-tocopherol for DPPH assay (EC50 = 22.86 µg/mL) and ascorbic acid for H2O2 assay (EC50 = 162.2 µg/mL).”. However, the blank controls of nanoemulsion without mangostin extracts were not prepared. As mentioned above, we think that the nanoemulsion was prepared using the same components and contents. So, the different results should come from the compounds in the extracts only. 

6. There are so many language errors in this manuscript. We recommend the authors asking a professional editor or native English speaker to fully improve the language of this manuscript.

Ans: The manuscript was carefully read for English by native speaker, Mr. Dave Chang from USA. 

Reviewer #2: The manuscript mainly presents a Techno-Biofunctionality of Mangostins Loaded Virgin Coconut Oil Nanoemulsion and Nanoemulgel. I have read this paper with great interest as I can see the value of the approach investigated by the authors. However, I have a few issues in some aspects:

The manuscript has a clear problem statement and solution, but it needs some revision for the English language.

Ans: Thank you very much. The English language was rechecked by native speaker Mr. Dave Chang from USA.

In fig.3 and fig.5, please indicate significant differences by different letters (statistical analysis).

Ans: The letters were added to indicate the statistical different in Fig. 3 and Fig. 5.

Could you explain more please results given in table 1.

Ans: The discussion regarding the result in Table 1 (which was now changed to Table 2) was originally detailed. We think that three pages discussion about this Table was enough and covered all aspects in the Table (droplet size, PDI and zeta potential). 

I would advise authors to complete the conclusion by adding the utility of this work.

Ans: The application of the mangostin extracts loaded virgin coconut oil nanoemulsion and nanoemulgel was recommended in the conclusion. 

Reviewer #3: In this manuscript named “Techno-Biofunctionality of Mangostins Loaded Virgin Coconut Oil Nanoemulsion and Nanoemulgel” Chatchai Sungpud et al are presenting a detailed technical and biofunctional comparison between Mangostins loaded nanoemulsions prepared from mangosteen peel extracts recovered by 3 different extractants (VCO, PG, and mixed VCO-PG). Based on an extensive characterization, the authors established that Mangostins loaded nanoemulsions prepared from mangosteen peel extracts recovered by mixed VCO-PG nanoemulsions were globular and uniformly distributed in nanoscale with the average droplet of less than 100 nm with negatively charged Zeta potentials of particles. The authors also tested the antioxidant and antibacterial activities of the nanoemulsions.

They also described the gel appearance, viscosity and mangostins release of mixed VCO-PG nanoemulgels as an excellent formulation for cosmetic based gel application.

Even though the manuscript is well written, nicely detailed and organized, I have few remarks:

*Abstract:

Modifications should be made to showcase the importance of your work and the potential of further interesting applications.

Less technical details and numbers should be included in the abstract.

Changes in the structure of the sentences should be introduced to make the abstract easier to follow.

Ans: The abstract was revised according to the reviewer’s suggestion.

*Material and Methods:

Nicely detailed techniques, explanations, and equations.

Since table 4 is the first table you refer to it, it should be named as table 1. 

Ans: Thank you very much. Table 4 was renamed as Table 1. Table 1, 2 and 3 were renamed as Table 2, 3 and 4, respectively.

*Results and discussion:

In “Stability of mangostins loaded nanoemulsions” paragraph, line 367, the authors described that freeze-thaw cycles affected the pH of the nanoemulsion and confirm that the final pH is still in the acceptable range of skincare (line 369), while the original pH is out of this range (less than 4).

Ans: Actually, the original pH is in the range of skin care. We have used the new specific reference and the value was updated to “3.5-6.0”. [Ali SM, Yosipovitch G. Skin pH: from basic science to basic skin care. Acta Derm Venereol.2013; 93:261-269.]

In “Antibacterial activities” paragraph, line 447, negative controls containing the extractant should be tested side by side with the corresponding nanoemulsions.

Ans: In this case, as stated in the material and method, we used DMSO as a negative control. Since each mangostin extract loaded nanoemulsion was diluted with DMSO before tested (P10-P11). 

Figure 1: The illustration of the panel with the same magnification is very important but it is very hard to see the nanoemulsion of PG extract prepared using HLB 15.1 (j) at this magnification. An additional picture with higher magnification should be added to proof that the nanoemulsions are present but in a smaller size.

Ans: The illustration of the panel with the same magnification is very important as reviewer stated. For the nanoemulsion of PG extract prepared using HLB 15.1, actually, the tiny white dots can be seen if zoomed in. You can recognize for all nanoemulsions with different extractants that the droplet sizes decreased with increasing HLB values. When it was printed out or zoomed in for the online PDF version, you can see the droplets. We would like to keep the same magnification for comparison. If the picture (j) was expanded, all pictures are needed too. So, the figures cannot be fitted in the same page.

A number of typos through the manuscript should be corrected.

Ans: All are corrected and the English was rechecked by native speaker, Mr. Dave Chang from USA. 

*Finally, this manuscript described an efficient strategy to prepare Mangostins loaded nanoemulsions prepared from mangosteen peel extracts with moderate particle size diameter scavenged more DPPH and H2O2 which can be used as a promising delivery system for bioactive polyphenol in food supplement, pharmaceutical and cosmetic. Furthermore, the VCO-PG nanoemulgel was classified as an excellent formulation for cosmetic based gel application.

Ans: We added the sentence in the conclusion to indicate the applicability of nanoemulsion produced. “Thus, nanoemulsions loaded with mangostin extracts can be used as a promising delivery system for bioactive polyphenol in food supplements, pharmaceuticals, and cosmetics.” 

Reviewer #4: Overall, this is a clear, concise, and well-written manuscript. The introduction is relevant and theory based. Sufficient information about the previous study findings is presented for readers to follow the present study rationale and procedures. The methods are generally appropriate, and well described. Overall, the results are clear and convincing. Overall, this is a high quality manuscript that has implications for the theoretical basis, devet.

Tables are cleae as well as figures.

This work is concidered for publication.

Ans: Thank you very much.

Ans: Done.

---

## [Decision Letter · Decision Letter 1]

6 Jan 2020

Techno-biofunctionality of mangostin extract-loaded virgin coconut oil nanoemulsion and nanoemulgel

PONE-D-19-30992R1

Dear Dr. Panpipat,

We are pleased to inform you that your manuscript has been judged scientifically suitable for publication and will be formally accepted for publication once it complies with all outstanding technical requirements.

With kind regards,

Walid Elfalleh, Ph.D

Academic Editor

PLOS ONE

Additional Editor Comments (optional):

The current version of the paper seems improved an all requested corrections were incorporated on the manuscript. I consider that the paper is worthy of publication.

Reviewers' comments:

Reviewer's Responses to Questions

**Comments to the Author**

1. If the authors have adequately addressed your comments raised in a previous round of review and you feel that this manuscript is now acceptable for publication, you may indicate that here to bypass the “Comments to the Author” section, enter your conflict of interest statement in the “Confidential to Editor” section, and submit your "Accept" recommendation.

Reviewer #1: All comments have been addressed

Reviewer #2: All comments have been addressed

Reviewer #3: All comments have been addressed

2. Is the manuscript technically sound, and do the data support the conclusions?

Reviewer #1: Yes

Reviewer #2: Yes

Reviewer #3: Yes

3. Has the statistical analysis been performed appropriately and rigorously? 

Reviewer #1: Yes

Reviewer #2: Yes

Reviewer #3: Yes

4. Have the authors made all data underlying the findings in their manuscript fully available?

Reviewer #1: Yes

Reviewer #2: Yes

Reviewer #3: Yes

5. Is the manuscript presented in an intelligible fashion and written in standard English?

Reviewer #1: Yes

Reviewer #2: Yes

Reviewer #3: Yes

6. Review Comments to the Author

Reviewer #1: The authors have addressed the reviewers' comments and improved the manuscript. Now it appears to be acceptable.

Reviewer #2: (No Response)

Reviewer #3: Chatchai Sungpud and al. did a very good job updating the abstract and going over the whole manuscript.

7. PLOS authors have the option to publish the peer review history of their article (what does this mean?). If published, this will include your full peer review and any attached files.

Reviewer #1: No

Reviewer #2: No

Reviewer #3: No

---

## [Editor Report · Acceptance letter]

9 Jan 2020

PONE-D-19-30992R1 

Techno-biofunctionality of mangostin extract-loaded virgin coconut oil nanoemulsion and nanoemulgel 

Dear Dr. Panpipat:

I am pleased to inform you that your manuscript has been deemed suitable for publication in PLOS ONE. Congratulations! Your manuscript is now with our production department. 

With kind regards,

on behalf of

Professor Walid Elfalleh 

Academic Editor

PLOS ONE